# Matching plasma and tissue miRNA expression analysis to detect viable ovarian germ cell tumors

Arkhjamil Angeles[1☯], Nastaran Khazamipour[2☯], Gurdial Dhillon[3], Sajjad Janfaza[2], Htoo Zarni Oo[2], Guliz Ozgun[1], Alireza Moeen[4,2], Craig Nichols[2,3,4], Christian Kollmannsberger[1], Mark S. Carey[1,3], Lucia Nappi [1,2]*

1 Department of Medicine, Medical Oncology Division, BC Cancer, Vancouver Centre, University of British Columbia, BC, Canada, 2 Department of Urologic Sciences, Vancouver Prostate Centre, University of British Columbia, Vancouver, BC, Canada, 3 Departments of Obstetrics and Gynecology and Clinical Research, University of British Columbia and BC Cancer, Vancouver, BC, Canada, 4 Testicular Cancer Commons, Beaverton, Oregon, United States of America

☯ These authors contributed equally to this work
* Lucia.Nappi@bccancer.bc.ca

## Abstract

### Purpose

MicroRNAs (miRNAs) are emerging as circulating biomarkers in germ cell tumors (GCT) with potential to guide management. Their role and expression patterns are more established in testicular GCTs, while lesser data exist in ovarian GCTs (OGCT).

### Methods

Patients diagnosed with OGCT with plasma and tumor tissue available in our provincial biobank were included. Total RNA was extracted, and RT-qPCR was performed to measure miR-371–3 and miR-302/367 levels. Healthy plasma and ovarian tissue served as controls. Statistical analyses were performed using ANOVA and the Mann-Whitney U test. Clinicopathologic data was collected by chart review.

### Results

From 2007 to 2022, 23 patients with OGCT were identified: 13 with viable non-teratoma germ cell (VNTGC) and 10 with immature teratoma germ cell (ITGC) tumors. Compared to healthy controls, all patients with VNTGC but not ITGC tumors had significantly higher miRNA levels in preoperative plasma and tumor tissue. Plasma miRNA kinetics correlated with disease burden, decreasing to undetectable levels following treatment, and increasing significantly upon relapse.

### Conclusion

MiR-371–3 and miR-302/367 are highly expressed in ovarian VNTGC but not ITGC tumors, and their plasma levels correlate with disease burden. Future studies

**Data availability statement:** All relevant data are within the manuscript and its Supporting Information files.

**Funding:** This study was funded by the Health Research BC-Health Professional Investigator Award given to Dr. Nappi. The funding body had no involvement in the study design, data collection or analysis, or decision to publish. The authors declare no other competing interests.

**Competing interests:** The authors declare no competing interests.

**Abbreviations:** GCT, germ cell tumor; OGCT, ovarian germ cell tumor; VNTGC, viable non-teratoma germ cell; ITGC, immature teratoma germ cell; YST, yolk sac tumor; EC, embryonal carcinoma; AFP, alpha-fetoprotein; HCG, human choriogonadotropin; LDH, lactate dehydrogenase; RT, reverse transcription; qPCR, quantitative polymerase chain reaction; Ct, cycle threshold; RQ, relative quantity; FFPE, formalin-fixed paraffin-embedded; ULN, upper limit of normal; BEP, bleomycin, etoposide, and platinum; POD, postoperative day; POY, postoperative year; PreOD, preoperative day.

validating these findings in a larger cohort are needed to develop miRNAs as circulating biomarkers for clinical use.

## Introduction

Ovarian germ cell tumors (OGCTs) are a heterogeneous group of malignancies that arise from the primordial germ cells of the ovary. They are histologically diverse, and can be classified as viable non-teratoma germ cell (VNTGC) tumors which include dysgerminomas, yolk sac tumors (YST), embryonal carcinomas, non-gestational choriocarcinomas, polyembryomas, and mixed germ cell tumors, or as mature and immature teratoma germ cell (ITGC) tumors [1]. They represent 5% of all ovarian neoplasms, and typically occur in younger women in contrast to the more common epithelial ovarian cancers [2,3].

The management of OGCTs has evolved over the past decades. Modeling after lessons learned in the more common testicular germ cell tumor (GCTs), standard management of OGCTs usually consists of primary diagnostic surgery with or without postoperative cisplatin-based chemotherapy [4]. Following primary treatment, the majority of patients are cured, but a proportion of patients will experience disease relapse. Accurate identification of patients at risk and earlier detection of disease recurrence are key to improve patient outcomes. Therefore, effective postoperative surveillance strategies are strongly recommended.

Current guidelines for surveillance endorse a program of routine clinical assessments, monitoring of serum tumor markers, and radiologic imaging [4]. However, these surveillance methods are often suboptimal. Clinical assessments are limited by a low sensitivity to identify asymptomatic early relapse [5,6]. Serum tumor markers – alpha-fetoprotein (AFP), human choriogonadotropin (HCG), lactate dehydrogenase (LDH), inhibin, and estradiol – can be unreliable as they are not frequently expressed by some OGCT histologies (such as dysgerminomas and teratomas) and can be abnormal in several non-malignant conditions such as inflammatory processes, substance use, and liver disease [7,8]. Further, standard radiologic imaging with computed tomography can be imprecise due to the use of nonspecific size criteria, and increase patient exposure to radiation [9–11]. Therefore, more accurate surveillance strategies are needed.

MicroRNAs are small non-coding RNAs that have shown promise as markers of malignant GCTs [12,13]. A family of miRNAs has been identified in GCTs with their serum levels correlating with disease burden and treatment response [14–20]. Specifically in testicular GCTs, studies have shown that miR-371 expression has a specificity, sensitivity, positive and negative predictive value of 100%, 96%, 100%, and 98%, respectively [19]. This is significantly more accurate in detecting active germ cell malignancy compared to standard surveillance strategies, and its accuracy has been demonstrated across a range of disease volumes from overt metastatic disease to equivocal disease such as clinical stage I disease, borderline enlarged retroperitoneal lymphadenopathy, and post-treatment residual disease [19]. Several large scale clinical trials are ongoing to validate the utility of this miRNA in order to

integrate it into clinical practice (S1823, NCT04435756; AGCT1531, NCT03067181; SWENOTECA MIR, NCT04914026; CLIMATE, ACTRN12622000247774).

In malignant OGCTs, few data have been reported. Chang et al. examined the miRNA expression profile in OGCT tissues and identified clusters of miRNAs that were overexpressed including miR-371–373 and miR-302/367 [20]. However, data validating the use of miRNAs as circulating biomarkers in OGCT are scarce, and studies often had limitations such as small patient numbers, inclusion of both testicular and ovarian GCTs, and few sample types and collection time points. Our study aims to evaluate the expression of these novel biomarkers in a population-based cohort of patients diagnosed with OGCT in both tumor tissue and plasma, and to monitor changes in levels during treatment and follow-up.

## Methods

### Patients and eligibility

Patients with malignant OGCT managed in British Columbia between January 1, 2007 and December 31, 2022, were eligible for this study. Patients must have adequate plasma and/or tumor tissue specimens accessible through the BC Gynecologic Oncology Biobank, which houses prospectively collected plasma and tumor tissues from patients diagnosed with gynecologic malignancy. Healthy control tissue and plasma samples were obtained with consent from women with non-malignant ovarian pathology. Clinicopathologic data and archived specimens were accessed between February 1, 2023, and August 31, 2023 for retrospective chart review and pathologic review, respectively. Authors had access to information that could identify individual participants during but not after the data collection for the purposes of performing a chart review. All patients provided written informed consent. The study was approved by the BC Cancer Agency Research Ethics Board (H22-03668-A001).

### MicroRNA extraction and quantification

**Plasma specimens.** Whole blood collected from patients with OGCT and healthy controls in Streck tubes (Streck, Cat# 230470) was centrifuged at 1,600 x g for 15 minutes at room temperature. The top layer was transferred to a new tube and centrifuged at 14,400 x g for 10 minutes at room temperature. The resulting plasma was aliquoted into separate tubes for processing. Total RNA was isolated from 100 μl of plasma using the miRNeasy Serum/Plasma Kit (QiAGEN, Hilden, Germany) according to the manufacturer's protocol. One μl of an external synthetic spike-in, cel-miR-39-3p, was added to each sample prior to extraction to ensure consistent RNA extraction quality across all samples. RNA concentration and purity were assessed using a NanoDrop spectrophotometer.

Primer pools were prepared for the multiplex reverse transcription (RT) and preamplification, including 10 primers of TaqMan MicroRNA Assay (hsa-miR-371-3p, ID: 002124; hsa-miR-372, ID:000560; hsa-miR-373, ID 000561; hsa-miR-367, ID: 000555; has-miR-302a, ID 000529; has-miR-302b, ID 000531; hsa-miR-302c, ID 000533; hsa-miR-302d, ID 000535; hsa-miR-30b, ID: 000602; cel-miR-39, ID: 000200). The primer pools were prepared in 1X TE buffer, so that the final concentration of each RT primer in the RT primer pool was 0.05X, and the final concentration of each preamplification primer in the preamplification primer pool was 0.2X. Multiplex RT was performed on 100 ng of RNA using the TaqMan MicroRNA Reverse Transcription Kit (Applied Biosystems, Waltham, MA, USA) followed by 12 cycles of multiplex preamplification on 5 ul of cDNA using the TaqMan PreAmp Master Mix Kit (Applied Biosystems, Waltham, MA, USA). All steps were performed according to the manufacturer's protocol. The final preamplification product was diluted 1:3 with RNase-free water. Levels of miR-302/367 and miR-371–373 clusters were measured using quantitative polymerase chain reaction (qPCR). Singleplex qPCR for individual miRNAs was performed in triplicate using the TaqMan miRNA Assay (Applied Biosystems, Waltham, MA, USA). Mean of cycle threshold (Ct) triplicates were normalized against the mean of the Ct value of the reference gene, miR-30b-5p, and calculated as ΔCt. The ΔCt values of the patient samples were normalized against the ΔCt values of the healthy controls, and the relative quantity (RQ) was expressed as $2^{-\Delta\Delta Ct}$, as previously reported [19].

**Tissue specimens.** For tumor tissue, microscope slides were prepared for each block and stained with hematoxylin-eosin. The tumor area was marked under the microscope, and tissue cores were harvested by inserting tissue microarray punch needles into regions of interest mapped onto formalin-fixed paraffin-embedded (FFPE) blocks. Two cores of 1–1.5 mm from each block were placed in a 1.5 mL Eppendorf tube. For healthy control tissue, two macro-dissected 10 μm sections from each FFPE block were placed into a 1.5 mL Eppendorf tube.

Both tumor and healthy control tissue samples were deparaffinized using CitriSolv (Decon Labs, Cat# 1601), and total RNA was extracted using the AllPrep DNA/RNA FFPE Kit (Qiagen, Hilden, Germany). The external synthetic spike-in, cel-miR-39-3p, was added to each sample after deparaffinization, to assess extraction efficiency. RNA concentration and purity were assessed using a NanoDrop spectrophotometer. RT, preamplification, and qPCR for miR-371–373 and miR-302/367 were performed as described above for the plasma specimens.

## Statistical analysis

Patient clinical, pathologic, and treatment characteristics were summarized using descriptive statistics. A comparison of ΔCt values between patients with OGCT and healthy controls was performed for both plasma (Fig 1A) and tissue (Fig 2A) specimens using two-way ANOVA followed by the Dunnett multiple comparison test. Relative quantification values of individual miRNAs for both plasma (Fig 1C) and tissue (Fig 2C) specimens were compared between VNTGC and ITGC tumors using the two-sided Mann-Whitney U test. Non-parametric tests were chosen to avoid assumptions about normal distribution of data. A p-value of $< 0.05$ was considered statistically significant.

The correlation coefficient (r) between plasma and tumor tissue miRNA level was calculated using non-parametric Spearman correlation in GraphPad Prism and visualized as a scatter plot (S1 Fig). P-values were adjusted using the Holm method (Adj p). The Concordance Correlation Coefficient (CCC) was calculated using the formula below:

$CCC = 2*Cov(X,Y)/\sigma X^2 + \sigma Y^2 + (\mu X - \mu Y)^2$, where:

- $\mu X$ and $\mu Y$ are the means of X (Tissue) and Y (Plasma)

- $\sigma X^2$ and $\sigma Y^2$ are the variances of X and Y

- $cov(X,Y)$ is the covariance between X and Y

To assess statistical differences between time points in patients 8 and 12, a one-way ANOVA test was performed separately for each miRNA. Each graph represents data from an individual patient with error bars indicating the mean of three technical replicates (S3 Fig).

## Results

### Patient characteristics

A total of 23 patients diagnosed with OGCT was identified in the BC Gynecologic Oncology Biobank (Table 1). The median age at diagnosis was 27 years (14–48). Histologically, 13 (57%) patients had VNTGC tumors, and 10 (43%) patients had ITGC tumors. Among patients with VNTGC tumors, 7 (30%) had pure dysgerminoma, 3 (13%) had mixed GCT with predominant YST histology, and 3 (13%) had pure YST. The median tumor size was 18.5 cm (9–35.5 cm). Most patients (61%) had International Federation of Gynecology and Obstetrics (FIGO) stage I disease where the primary tumor is limited to the ovaries or fallopian tubes at initial diagnosis.

Traditional tumor markers at diagnosis and post initial treatment that were available are summarized in Table 1. At diagnosis, HCG was above the upper limit of normal (ULN) in 11 of 22 patients. Specifically, all patients with dysgerminomas and mixed GCT with predominant YST histology had an abnormal HCG, while none of the patients with immature teratoma except one had an abnormal HCG. LDH at diagnosis was above the ULN in 10 of 22 patients. Specifically, all patients with dysgerminoma had an LDH above the ULN, while 1 of 2 patients with mixed GCT with predominant YST

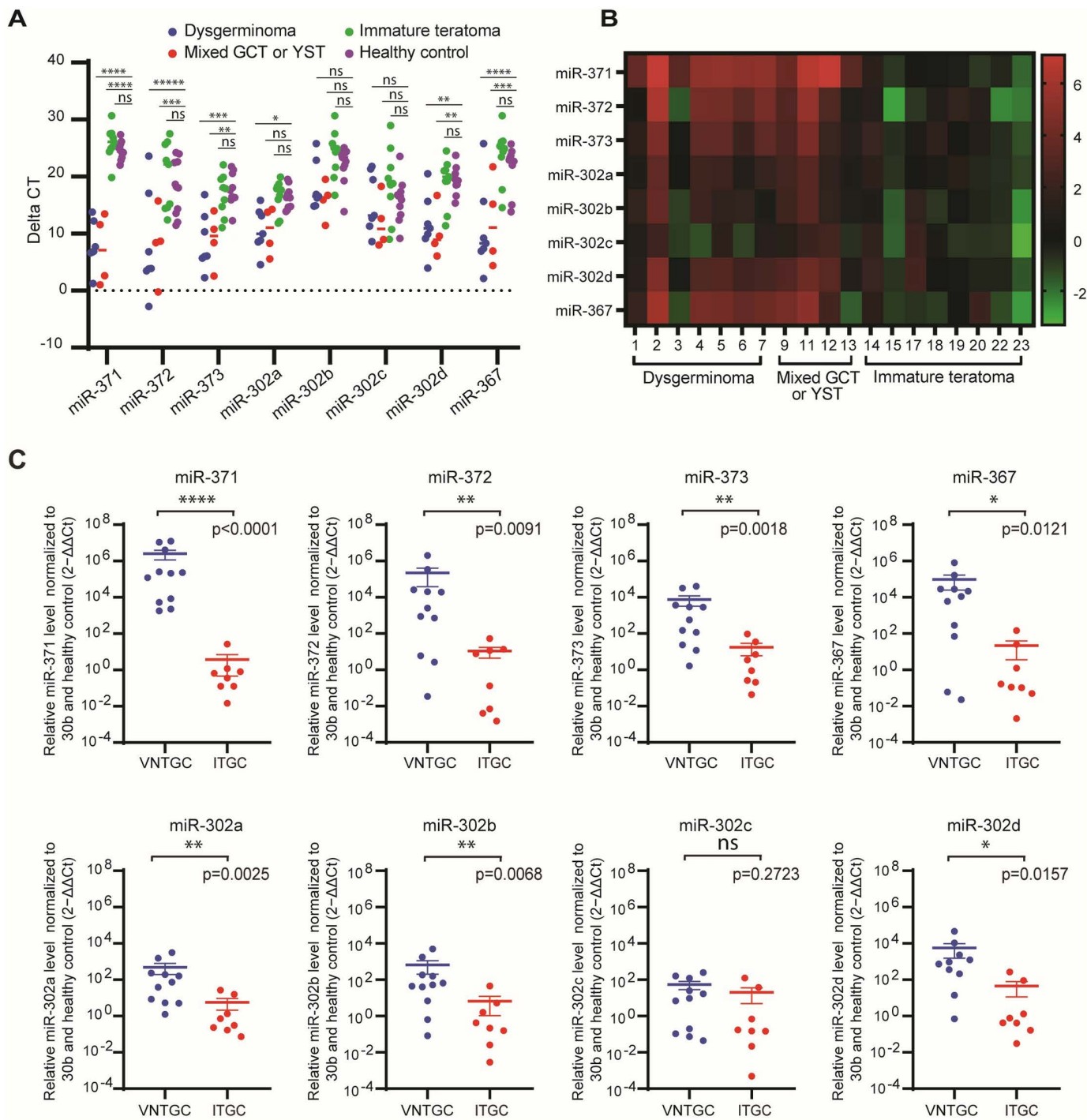

**Fig 1. Expression analyses of two clusters of miRNA (371-3 and 302a-d/367) in the preoperative plasma of patients with ovarian germ cell tumor.** A) Plot comparison of individual miRNA expression levels for each type of ovarian germ cell tumor and for healthy controls. B) Heatmap of RQ values of individual miRNA expression for each patient. Patient IDs are identical to those in Table 1. C) Plot comparisons of the RQ values of individual miRNA expression levels for VNTGC and ITGC tumor groups. miRNA expression was quantified as fold of expression increase relative to the healthy controls, after normalization for miR-30b-5p, and presented as 2-ΔΔCt. The statistical significance between VNTGC and ITGC tumors was determined using two-sided Mann-Whitney U test. *, p, <0.05; **, p, <0.01; ***, p, <0.001; ****, p, <0.0001. The Y axis is plotted on a log10 scale. GCT, germ cell tumor; YST, yolk sac tumor; Ct, cycle threshold; VNTGC, viable non-teratoma germ cell; ITGC, immature teratoma germ cell.

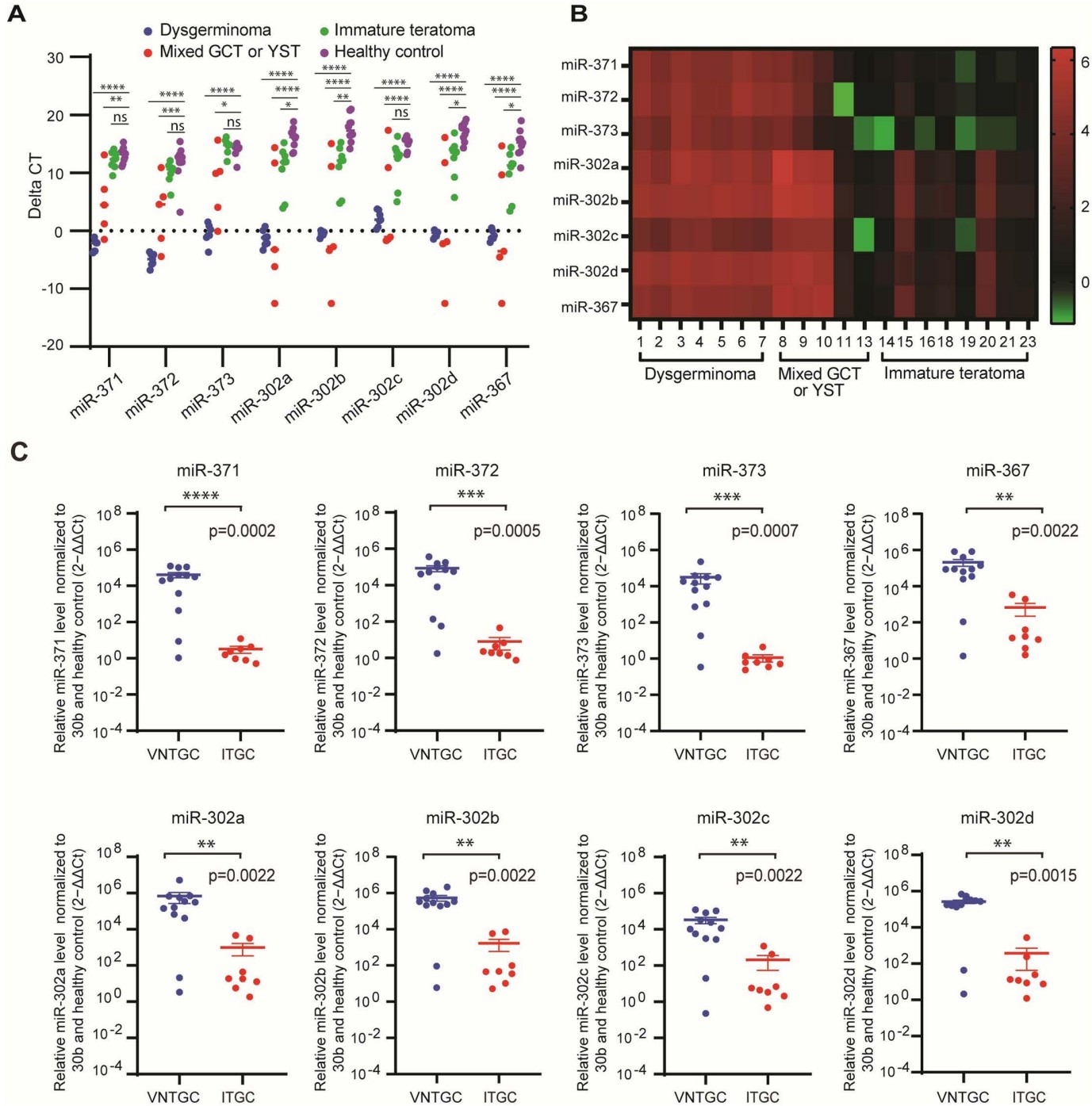

**Fig 2. Expression analyses of two clusters of miRNA (371-3 and 302a-d/367) in the surgical tumor tissue of patients with ovarian germ cell tumor.** A) Plot comparison of individual miRNA expression levels for each type of ovarian germ cell tumor and for healthy controls. B) Heatmap of RQ values of individual miRNA expression for each patient. Patient IDs are identical to those in Table 1. C) Plot comparisons of the RQ values of individual miRNA expression levels for VNTGC and ITGC tumor groups. miRNA expression was quantified as a fold of expression increase relative to the healthy controls after normalization for miR-30b-5p, and presented as 2-ΔΔCt. The statistical significance between VNTGC and ITGC tumors was determined using a two-sided Mann-Whitney U test. \*, p, < 0.05; \*\*, p, < 0.01; \*\*\*, p, < 0.001; \*\*\*\*, p, < 0.0001. The Y axis is plotted on a log10 scale. GCT, germ cell tumor; YST, yolk sac tumor; Ct, cycle threshold; VNTGC, viable non-teratoma germ cell; ITGC, immature teratoma germ cell.

**Table 1.** Clinicopathologic characteristics of 23 patients with ovarian germ cell tumors.

| Patient ID | Age (years) | Histological type | Tumor size (cm) | Initial FIGO stage | HCG (IU/L) | | LDH (U/L) | | AFP (mg/L) | |
|---|---|---|---|---|---|---|---|---|---|---|
| | | | | | Pre-surgery | Post-surgery | Pre-surgery | Post-surgery | Pre-surgery | Post-surgery |
| 1 | 26 | Dysgerminoma | 15 | IIIC | + | – | + | – | – | – |
| 2 | 16 | Dysgerminoma | 29* | IVB | + | – | + | – | – | na |
| 3 | 17 | Dysgerminoma | 13* | IA | + | – | + | – | – | – |
| 4 | 17 | Dysgerminoma | na | IIIC | + | – | + | – | – | – |
| 5 | 20 | Dysgerminoma | 33 | IA | + | – | + | – | – | – |
| 6 | 41 | Dysgerminoma | 23 | IIIC | + | – | + | – | – | na |
| 7 | 22 | Dysgerminoma | 23 | IC | + | – | + | – | – | – |
| 8 | 18 | Mixed GCT (85% YST, 15% mature teratoma) | 21 | IIIA | + | – | – | – | + | – |
| 9 | 30 | Mixed GCT (80% YST, 10% dysgerminoma, 10% immature teratoma) | 16 | IC | + | – | na | – | + | – |
| 10 | 48 | Mixed GCT (YST, EC, immature teratoma) | 35.5 | IC | + | – | + | – | na | – |
| 11 | 21 | YST | 9* | IA | – | na | – | na | + | na |
| 12 | 30 | YST | 15* | IC | – | – | + | – | + | – |
| 13 | 31 | YST | 17.5 | IIIC | – | – | – | – | + | – |
| 14 | 30 | Immature teratoma | 18 | IIIA | – | – | – | – | + | – |
| 15 | 28 | Immature teratoma | 17 | IC | + | – | – | na | + | – |
| 16 | 27 | Immature teratoma | 16* | IC | – | – | – | – | + | – |
| 17 | 14 | Immature teratoma | 24* | IA | – | – | + | na | na | – |
| 18 | 23 | Immature teratoma | 22 | IA | – | na | – | – | + | – |
| 19 | 31 | Immature teratoma | 11* | IIIA | – | – | – | – | + | – |
| 20 | 30 | Immature teratoma | 25 | IA | – | – | – | na | + | – |
| 21 | 31 | Immature teratoma | 30 | IC | na | – | – | – | + | – |
| 22 | 18 | Immature teratoma | 13* | IC | – | – | – | – | + | – |
| 23 | 45 | Immature teratoma | 19 | IIB | – | na | – | na | + | – |

*Gross pathologic size

GCT, germ cell tumor; YST, yolk sac tumor; EC, embryonal carcinoma; FIGO, International Federation of Gynecology and Obstetrics; HCG, human choriogonadotropin; LDH, lactate dehydrogenase; AFP, alpha-fetoprotein; +, above upper limit of normal; -, normal; na, not available.

histology, 1 of 2 patients with YST, and 1 of 10 patients with immature teratoma had an LDH above the ULN. AFP at diagnosis was above the ULN in 14 of 21 patients. Specifically, all patients except those with dysgerminoma histology had an abnormal AFP. Serum tumor markers normalized post-treatment in all but one patient with immature teratoma who had a reduced but persistently abnormal AFP. She was treated with cytoreductive surgery and postoperative chemotherapy, and later was found to have recurrent disease in the pelvis over 3 years after completing initial treatment.

### miRNA expression in preoperative plasma

Nineteen of the 23 patients with OGCT had preoperative plasma available for evaluation. Most patients had plasma collected on the day of their surgery (day -27–0).

A comparison of the ΔCt and median RQ values for miR-302/367 and miR-371–373 in the preoperative plasma of patients with OGCT and healthy controls is summarized in Fig 1A and 1B. Of the 7 patients with pure dysgerminoma, 7 had preoperative plasma for evaluation, and they all demonstrated higher levels of expression of all miRNAs when

compared to healthy controls. Of the 6 patients with YST or mixed GCT with predominant YST histology, 4 had preoperative plasma for evaluation and all four samples demonstrated higher levels of expression of all miRNAs when compared to healthy controls. Of the 10 patients with immature teratoma, 8 had preoperative plasma for evaluation, and all eight samples demonstrated similar levels of expression of all miRNAs when compared to healthy controls (Fig 1A and 1B).

A comparison of RQ values for miR-302–367 and miR-371–373 in the preoperative plasma of patients with VNTGC and ITGC tumors is illustrated in Fig 1C. Patients with VNTGC compared to ITGC tumors had significantly higher expression levels of all miRNAs except miR-302c in preoperative plasma: miR-371a ($p < 0.0001$), miR-372 ($p = 0.0091$), miR-373 ($p = 0.0018$), miR-367 ($p = 0.0121$), miR-302a ($p = 0.0025$), miR-302b ($p = 0.0068$), miR-302c ($p = 0.2723$), and miR-302d ($p = 0.0157$).

## miRNA expression in tumor tissue

Of the 23 patients with OGCT, three patients (ID #12, #17, and #22) did not have sufficient tumor tissue available for downstream processing and RNA quantification.

A comparison of the ΔCt and median RQ values for miR-302–367 and miR-371–373 in the tumor tissue of patients with OGCT and healthy controls is summarized in Fig 2A and 2B. All 7 patients with pure dysgerminoma showed significantly higher expression levels of all miRNAs when compared to healthy tissue controls ($p < 0.0001$). This was fully concordant with the preoperative plasma results (Figs 3 and S1). The 3 patients with mixed GCT with predominant YST histology showed significantly higher levels of expression of all miRNAs. Only one patient (ID #9) had preoperative plasma available for comparison, and this was concordant with tumor tissue expression. Two of the three patients with YST had tumor tissue available, and they showed similar expression levels of all miRNAs when compared to healthy tissue controls. This was discordant with preoperative plasma results that showed elevated miRNA expression, which was also observed in the patient (ID #12) without evaluable tumor tissue.

Eight out of ten patients with immature teratoma had tumor tissue available for evaluation. Of the 8 patients, 6 patients showed no significant miRNA overexpression compared to healthy tissue controls. Four of the six patients had preoperative plasma for comparison, and all 4 patients showed concordant preoperative plasma results with no significant miRNA expression when compared to healthy controls. Two of the 8 patients with ITGC tumor tissue showed significant miRNA overexpression for miR302a/b/d and miR367 compared to healthy tissue controls, and this was discordant with their preoperative plasma results which showed no significant miRNA expression. Two of the ten patients with ITGC tumor tissue (ID #16 and #21) did not have preoperative plasma available for comparison. For the 2 patients (ID #17 and #22) reported to have immature teratoma on initial pathologic review but with no tumor tissue available for evaluation, both patients had similar miRNA expression levels when compared to healthy controls.

A comparison of RQ values for individual miRNAs levels in the tumor tissue of patients with VNTGC and ITGC tumors is illustrated in Fig 2C. Patients with VNTGC tumor had significantly higher expression levels of all miRNAs in tumor tissue compared to those with ITGC tumors: miR-371a ($p = 0.0002$), miR-372 ($p = 0.0005$), miR-373 ($p = 0.0007$), miR-367 ($p = 0.0022$), miR-302a ($p = 0.0022$), miR-302b ($p = 0.0022$), miR-302c ($p = 0.0022$), and miR-302d ($p = 0.0015$).

## miRNA expression in postoperative plasma

Four of the 23 patients with OGCT had postoperative plasma samples available for evaluation.

None of the patients with dysgerminoma had postoperative plasma samples. One of the 3 patients with mixed GCT with predominant YST histology (ID #8) had postoperative plasma samples. She had cytoreductive surgery with large volume residual disease. Her plasma sample 5 days after her initial surgery showed detectable but low miRNA expression levels. She did not have preoperative plasma available for comparison. She received 4 cycles of postoperative chemotherapy with bleomycin, etoposide, and platinum (BEP) starting 5 days and completing 66 days after her initial surgery. Three plasma samples were taken while she was on chemotherapy which showed declining miRNA levels that eventually

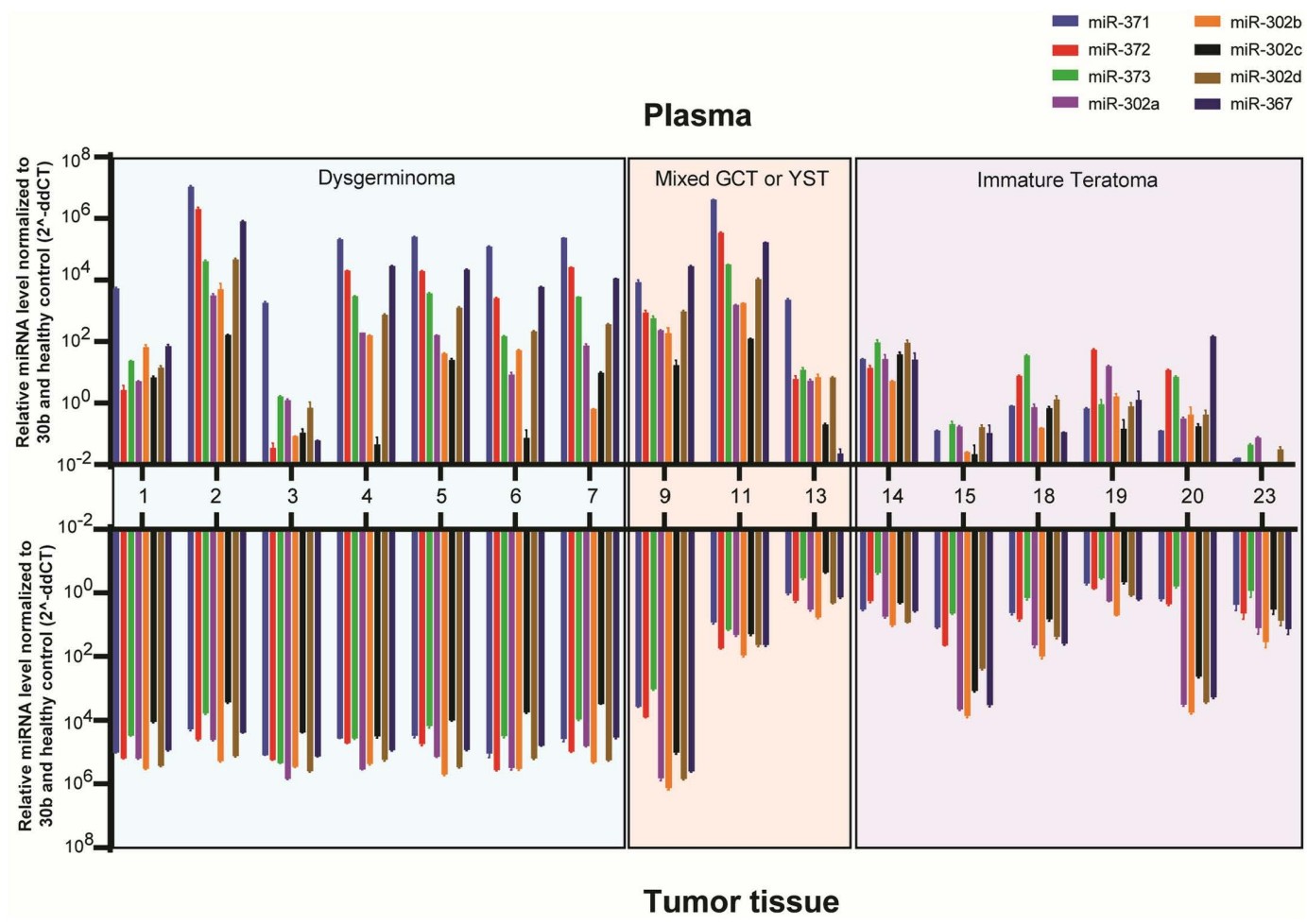

**Fig 3. Comparison of individual miRNA expression levels between preoperative plasma and tumor tissue samples for each patient with ovarian germ cell tumor (only 17 patients had both samples available for analysis).** The Y axis is plotted on a log10 scale. Patient IDs are identical to those in Table 1.

became undetectable near completion of her systemic treatment. She developed recurrent disease in the pelvis detected on CT imaging about 5 years after completing postoperative chemotherapy. Plasma collected at the time of recurrence showed recurrently elevated miRNA levels (Figs 4A and S3A).

One of the 3 patients with YST (ID #12) had postoperative plasma samples. Her plasma miRNA levels 35 days after her initial surgery reduced to undetectable levels compared to preoperative levels. She proceeded to have postoperative chemotherapy with 3 cycles of BEP. Almost 4 years after completing chemotherapy, she developed recurrence in the pelvis detected on CT imaging. Plasma samples collected at the time of recurrence showed recurrently elevated miRNA levels (Figs 4B and S3B).

Two of the 10 patients (ID #16 and #21) with immature teratoma had postoperative plasma samples, and both showed no significant miRNA overexpression after cytoreductive surgery and postoperative chemotherapy with BEP compared to healthy plasma controls (Fig 5). Both patients similarly had no significant preoperative plasma miRNA expression.

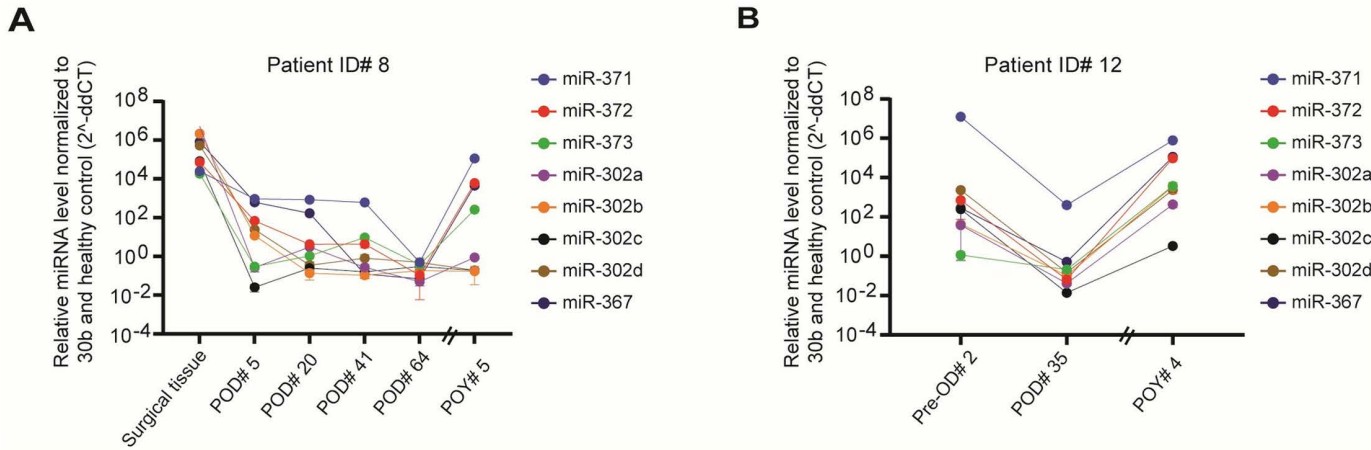

**Fig 4. Serial measurement of miRNA expression from the time of diagnosis to recurrence in two patients with viable non-teratoma germ cell tumors.** A) RQ values of individual miRNA expression levels of surgical tissue and postoperative plasma samples over time for patient ID #8 with mixed GCT with predominant YST histology. Samples include surgical tumor tissue, POD# 5 plasma (with postoperative large volume residual disease), POD# 20, 41, and 64 plasma after 1, 2, and 4 cycles of BEP chemotherapy, respectively, and POY# 5 plasma at the time of disease recurrence. B) RQ values of individual miRNA expression levels of preoperative and postoperative plasma samples over time for patient ID# 12 with pure YST histology. Samples include PreOD# 2 plasma, POD# 35 plasma, and POY# 4 plasma at the time of disease recurrence. The Y axis is plotted on a log10 scale. RQ, relative quantity; GCT, germ cell tumor; YST, yolk sac tumor; POD, postoperative day; PreOD, preoperative day; POY, postoperative year; BEP, bleomycin, etoposide, and platinum. Error bars represent mean ± SEM of three technical replicates.

## Discussion

In our study, we showed that the expression of two clusters of miRNAs, miR-302/367 and miR-371–3, were significantly elevated and concordant in both the preoperative plasma and tumor tissue of nearly all patients with VNTGC tumors. This is consistent with other studies that have shown elevated expression of miRNA in various OGCT specimens including tumor tissue, pre-treatment plasma, and other bodily fluids including cerebrospinal fluid in patients with primary central nervous system GCT [20,21]. The only exceptions were patients with pure YST histology who had discordant miRNA expression showing significantly elevated miRNA levels in preoperative plasma but not in tumor tissue. This is different from what has been reported for testicular and ovarian GCTs [12,13,18,22], and one of the reasons for this discordance may be related to low tumor tissue sample yield in some of the YST specimens.

We also observed differential miRNA expression in the preoperative plasma of VNTGC tumors with miR-371 being the most highly expressed among the two miRNA clusters. This is consistent with other reports including in patients with testicular and ovarian GCT [15,16,20], and may highlight miR-371 as being the best candidate biomarker for OGCT. In contrast, expression levels among the two miRNA clusters appeared more similar in tumor tissue. Many reasons may account for this difference including potentially higher stability of miR-371 in plasma which has been demonstrated in some studies [23], or greater quantities of extra-tumoral release of miR-371 relative to other miRNA species.

In contrast, the two miRNA clusters were not significantly elevated in all preoperative plasma and most of the tumor tissue of patients with ITGC tumors. This is consistent with the miRNA expression profile analysis in benign OGCTs by Chang et al. and with the data reported in testicular teratoma [20]. Only two patients with ITGC had tumor tissue that showed significant expression in miR302a/b/d and miR367. This may suggest that these specific clusters of miRNA may be less specific for ITGC tumors. Additionally, other methods of evaluation may be more relevant in this specific GCT histology such as the integrated measurement of miR-375-miR-371, which was shown to be more accurate than individual miRNAs in detecting testicular teratomas [24].

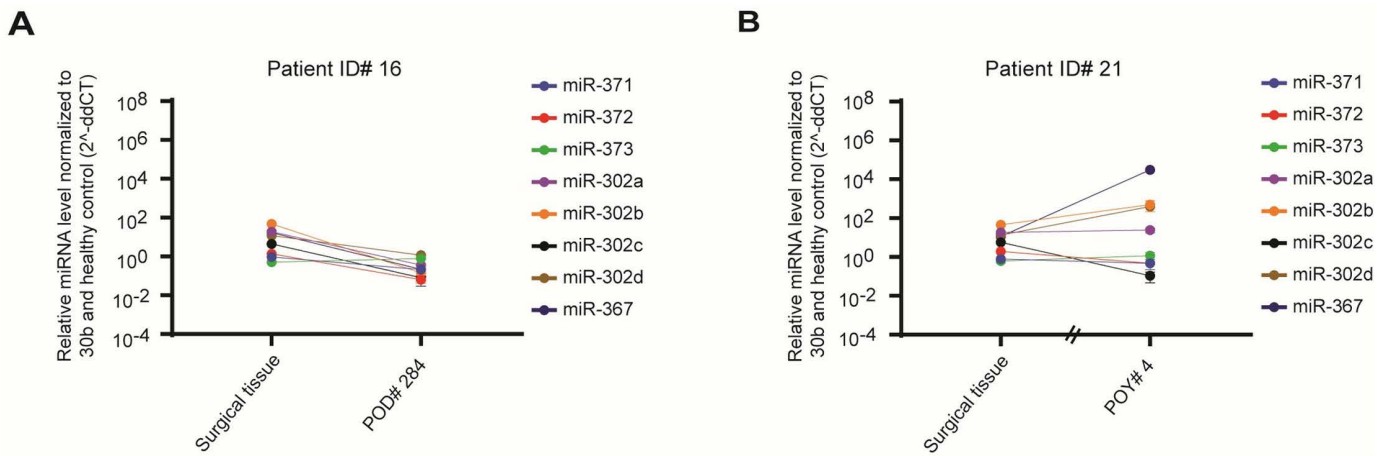

**Fig 5. Measurement of miRNA expression in surgical tissue and postoperative plasma in two patients with immature teratoma germ cell tumors.** A) RQ values of individual miRNA expression levels of surgical tissue and POD# 284 plasma for patient ID #16 with mixed teratoma histology. B) RQ values of individual miRNA expression levels of surgical tissue and POY# 4 plasma for patient ID #21 with teratoma histology. The Y axis is plotted on a log10 scale. RQ, relative quantity; POD, postoperative day; POY, postoperative year. Error bars represent mean±SEM of three technical replicates.

In the two patients with VNTGC tumors with serial plasma samples available for analysis, we showed that the kinetics of miRNA levels in plasma appear to correlate with disease burden and treatment with declining miRNA levels seen postoperatively that eventually reach normal control levels after the completion of postoperative systemic therapy. Similar miRNA kinetics following treatment have been observed in testicular GCTs [18,25,26]. However, in these two patients, reaching undetectable post-treatment levels did not predict the likelihood of cure as both patients eventually experienced disease recurrence years after completing initial therapy. At the time of disease recurrence, plasma miRNA levels became detectable again suggesting potential diagnostic value during longitudinal follow-up.

We were not able to make formal comparisons between the operating characteristics (i.e., sensitivity, specificity, predictive value) of circulating miRNAs and traditional serum tumor markers in OGCTs because only a few patients had serial plasma samples available for miRNA expression quantification and analysis. For our cohort of 13 VNTGC tumors, their preoperative plasma were all consistently positive for elevated miRNA expression while their preoperative traditional tumor markers were less consistent. For instance, HCG was abnormal in patients with dysgerminoma and mixed GCT with predominant YST, but negative in pure YST. Also, LDH was abnormal in some cases of mixed GCT and YST, but normal in others. This is not unexpected as traditional tumor markers have repeatedly been reported to lack sensitivity and specificity which limit their use [7,8]. It is possible that miRNAs have similar pitfalls as well as some studies suggest elevated circulating miR-371–3 levels in patients with thyroid neoplasms [27]. As such, further work that characterize the operating characteristics of circulating miRNAs in different disease contexts are needed to better understand their potential role as diagnostic markers.

Several limitations may impact the interpretation of the study results. Although our study, to our knowledge, represents one of the largest patient cohorts examining miRNA expression over time in patients with OGCT, the absolute patient sample size was still small especially when further subclassification by histology is done for data analysis. It is therefore uncertain if our findings will hold true in a larger patient cohort. Additionally, patient serum samples in the postoperative setting were scarce, and when available, were taken at irregular time points. This makes it difficult to fully understand plasma miRNA kinetics and operating characteristics compared to standard surveillance methods.

Our study adds to the small but growing body of evidence that supports the potential role of circulating miRNAs as clinical biomarkers in OGCT. They appear to hold greater diagnostic value than traditional tumor markers. But more work needs to be done before circulating miRNAs can be used to help guide routine OGCT managementbe including gaining a better understanding of their operating characteristics in various stages of disease, and determining their role in guiding clinical decision making in settings such as postoperative therapy and surveillance. Large transnational collaborations to define clinical utility and impact a highly specific biomarker on surveillance and quality of survivorship are in discussion.

## Conclusion

MiR-371–3 and miR-302/367 are highly expressed in ovarian VNTGC tumor tissue and plasma. Their plasma levels on serial measurement correlate with changing disease burden. Although these findings are based on a limited retrospective patient cohort, they suggest a potential role for miRNAs as clinical biomarkers in OGCT.

## Supporting information

**S1 Fig. Correlation of individual miRNA expression between plasma and tumor tissue.** Red dots represent patients with ITGC tumors, and blue dots represent VNTGC tumors. The Correlation Coefficient (r) was calculated in Graphpad Prism, and the Concordance Correlation Coefficient (CCC) was determined as described in statistical analysis section. (TIF)

**S2 Fig. Expression levels of selected miRNAs across different germ cell tumor types.** Bar graphs represent relative miRNA expression in teratoma, dysgerminoma, and mixed GCT in both plasma (top) and tumor tissue (bottom). Statistical comparisons were performed using one-way ANOVA followed by Dunnett's post-hoc test to identify differential expression of miRNAs in specific tumor types. (TIF)

**S3 Fig. Comparison of miRNA levels at different time points for each miRNA in patient IDs (A) #8 and (B) #12.** Statistical differences were analyzed using a one-way ANOVA test. Each graph represents data from an individual patient with bars indicating the mean of three technical replicates. (TIF)

## Acknowledgments

None

## Author contributions

**Conceptualization:** Nastaran Khazamipour, Arkhjamil Angeles, Lucia Nappi.

**Data curation:** Nastaran Khazamipour, Arkhjamil Angeles, Gurdial Dhillon, Sajjad Janfaza.

**Formal analysis:** Nastaran Khazamipour, Gurdial Dhillon, Sajjad Janfaza.

**Funding acquisition:** Arkhjamil Angeles, Christian Kollmannsberger, Lucia Nappi.

**Investigation:** Nastaran Khazamipour, Arkhjamil Angeles, Guliz Ozgun.

**Methodology:** Nastaran Khazamipour, Htoo Zarni Oo, Alireza Moeen.

**Project administration:** Nastaran Khazamipour.

**Software:** Nastaran Khazamipour.

**Supervision:** Craig Nichols, Christian Kollmannsberger, Mark S. Carey, Lucia Nappi.

**Validation:** Nastaran Khazamipour.

**Visualization:** Nastaran Khazamipour.

**Writing – original draft:** Nastaran Khazamipour, Arkhjamil Angeles.

**Writing – review & editing:** Nastaran Khazamipour, Arkhjamil Angeles, Gurdial Dhillon, Sajjad Janfaza, Htoo Zarni Oo, Alireza Moeen, Craig Nichols, Christian Kollmannsberger, Mark S. Carey, Lucia Nappi.

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
