## [Decision Letter · Decision Letter 0]

7 Jan 2025

PONE-D-24-42448Matching plasma and tissue miRNA expression analysis to detect viable ovarian germ cell tumorsPLOS ONE

Dear Dr. Nappi,

Thank you for submitting your manuscript to PLOS ONE. After careful consideration, we feel that it has merit but does not fully meet PLOS ONE’s publication criteria as it currently stands. Therefore, we invite you to submit a revised version of the manuscript that addresses the points raised during the review process.

We look forward to receiving your revised manuscript.

Kind regards,

Julie Decock, PhD

Academic Editor

PLOS ONE

Journal Requirements:

3. We note that your Data Availability Statement is currently as follows: “All relevant data are within the manuscript and in Supporting Information files.”

6. Please include your tables as part of your main manuscript and remove the individual files. Please note that supplementary tables (should remain/ be uploaded) as separate "supporting information" files.

Reviewers' comments:

Reviewer's Responses to Questions

**Comments to the Author**

1. Is the manuscript technically sound, and do the data support the conclusions?

Reviewer #1: Yes

Reviewer #2: Yes

Reviewer #3: Yes

2. Has the statistical analysis been performed appropriately and rigorously? 

Reviewer #1: Yes

Reviewer #2: Yes

Reviewer #3: Yes

3. Have the authors made all data underlying the findings in their manuscript fully available?

Reviewer #1: Yes

Reviewer #2: Yes

Reviewer #3: Yes

4. Is the manuscript presented in an intelligible fashion and written in standard English?

Reviewer #1: Yes

Reviewer #2: Yes

Reviewer #3: No

5. Review Comments to the Author

Reviewer #1: Introduction

Please define at least once the acronyms used in the introduction.

Methods

Line 140: Provide the manufacturer and catalog number for the Streck tubes used in the study.

Line 141: Instead of using the term ‘in a standard fashion’, please provide the exact conditions for the centrifugation, e.g., 5,000 rpm for 10 minutes at 4 °C.

Results

Line 204: Please explain the definition of FIGO and include a reference.

Table 1: Please include the definitions of the acronyms at the bottom of the table for better interpretation.

Figure 3: Instead of showing the results in graphs for visual comparison, consider plotting correlation graphs to reach miRNA so that this correlation can be quantifiable.

Figure 4: Have you done statistical analysis comparing the time points assessed? That would be important to show in the graph. Especially the comparison of the first point (Surgical tissue, Pre-OD# 2) to the other time points.

Reviewer #2: Angeles et al present data obtained from 23 patients that have ovarian germ cell tumors which reveals increased expression of miR-371-3p, as well as miR-302/367, within plasma and tumor tissue samples. These miRNAs are known to have higher expression within testicular germ cell tumors. They show increases in the incidence of these miRNAs prior to tumor removal, decreases in their expression after their removal, and return of high expression after a given amount of time after removal. These are all strong evidences that the expression of these miRNAs are, at the very least, correlated with the presence of malignant tumors that originated as OGCT. The manuscript does a good job of transparently presenting their supporting data, however, there are a few points of criticism/potential improvement that need addressing:

1) Lack of data showing relation to mRNA targets of miR-371-3/302/367

a. The evidence of these miRNAs presence and overexpression in OGCT patients, decrease after surgery, and re-increase well-after resection is very well-established in the paper. However, miRNAs do not act alone to cause oncogenicity, and therefore, it would be further solidifying of the authors’ data to show inverse patterns of known targets of these miRNA such as LATS2, TGFBR2, CDKN1A, or PIK3R1 in each of the patients shown in Figures 4 and 5. Although the authors’ intention was not to provide evidence for a putative pathway, it would certainly add to the significance of their findings. To even further concretize the authors' findings, they could create a ceRNA network related to all miRNA in the paper to show possible avenues of action instead of simple correlations.

2) Data-driven comparison of OGCT to TGCT

a. The authors’ rely on the similarity of testicular germ cell tumors within the introduction, and discussion, sections but do not provide any comparative data. Considering that the TGCT are the more well-established tumor to miRNA dysregulation, a comparison to currently available data from theses tumors would further solidify the connection between these primordial sex cells. Additionally, it could provide avenues for further investigation between to the two to find similarities, differences, and correlations.

3) Addressing Tumor Heterogeneity

a. Within the authors’ cohort, there is a wide-range of expression levels of mentioned miRNAs between patients (Figure 3). It would be helpful to delineate in a supplementary and separate figure which miRNAs are most overrepresented in specific tumor types as addressed in the discussion section. It’s one thing to recognize the difference in expression between tumor types, but it’s another, more useful, thing to present those differences quantitatively in graphical form. This could also add to the conclusion that germ cell tumors express specific miRNAs, regardless of sex type.

Overall, this paper contains a good amount of well-analyzed data that could be amplified with minor additions of supporting data as well as the inclusion of new figures that would help clearly show the conclusions of the authors.

Reviewer #3: The manuscript “Matching plasma and tissue miRNA expression analysis to detect viable ovarian germ

cell tumors” by Angeles et. al., investigated role and expression pattern of microRNAs (miRNAs) in ovarian GCTs

(OGCT).

Authors needs to address the following concerns to improve the quality:

1. Authors pointed out abnormal LDH while presenting results. It should be either higher than or lower than normal.

2. Line 212: "AFP at diagnosis was above ULN in 14 of 22 patients." Table 1 says 21 patients, please verify. Authors should consider presenting "Patient Characteristics" section with more clear statements.

3. Authors should use better resolution image for Fig 1 and 2, where labels are almost unreadable.

4. Please point out figure number and sub figure number clearly in the result section for better readability.

5. Conclusion should specifically point out the limitations of this study.

6. PLOS authors have the option to publish the peer review history of their article (what does this mean? ). If published, this will include your full peer review and any attached files.

**Do you want your identity to be public for this peer review?** For information about this choice, including consent withdrawal, please see our Privacy Policy .

Reviewer #1: No

Reviewer #2: **Yes: ** Brian Jorgensen

Reviewer #3: No

---

## [Author Response · Author response to Decision Letter 1]

26 Feb 2025

Reviewer #1

Introduction

1. Please define at least once the acronyms used in the introduction.

● Response: All acronyms, including FIGO, OGCT, and miRNAs, have now been defined at their first mention in the Introduction.

(OGCTs ) defined in Page 5 line 75

viable non-teratoma germ cell (VNTGC): Page 5 line 75

immature teratoma germ cell (ITGC): Page 5 line 77

germ cell tumor (GCTs): Page 5 line 82

Methods

2. Line 140: Provide the manufacturer and catalog number for the Streck tubes used in the study.

● Response: We have included the manufacturer (Streck, Cat#: 230470) in the Methods section. Page 8 line 143

3. Line 141: Instead of using the term ‘in a standard fashion,’ please provide the exact conditions for the centrifugation.

● Response: The specific conditions for centrifugation have now been added.

It is changed to ‘’ was centrifuged at 1,600 x g for 15 minutes at room temperature. The top layer was transferred to a new tube and centrifuged at 14,400 x g for 10 minutes at room temperature’’. Page 8 line 143

Results

4. Line 204: Please explain the definition of FIGO and include a reference.

● Response: We have included the explanation as follows. “Most patients (61%) had International Federation of Gynecology and Obstetrics (FIGO) stage I disease where the primary tumor is limited to the ovaries or fallopian tubes at initial diagnosis.’’ Page 11 line 219-220

5. Table 1: Please include the definitions of the acronyms at the bottom of the table for better interpretation.

● Response: All acronyms in Table 1 are already defined in the bottom of the table.

6. Figure 3: Instead of showing the results in graphs for visual comparison, consider plotting correlation graphs to reach miRNA so that this correlation can be quantifiable.

● Response: Thank you for your valuable suggestion. We have created a supplementary (Fig.S1) figure that includes separate correlation graphs for each miRNA, illustrating the relationship between miRNA levels in plasma and tumor tissue. Additionally, we have calculated the concordance correlation coefficient (CCC) for each graph to provide a quantifiable measure of correlation.

7. Figure 4: Have you done statistical analysis comparing the time points assessed? That would be important to show in the graph. Especially the comparison of the first point (Surgical tissue, Pre-OD# 2) to the other time points.

● Response: Thank you for your insightful suggestion. To assess whether the changes in miRNA expression at different timepoints were statistically significant, we conducted an ANOVA test for each miRNA. The results, along with separate graphs for each miRNA, have been included in a supplementary figure (see supplementary Fig 3) to provide a clear and comprehensive representation of the statistical significance of the observed trends. Since each graph in Figure 4 represents an individual patient, we performed the statistical analysis using technical replicates. The statistical analysis are explained in page 11, line 206-208..

Reviewer #2

1. Lack of data showing relation to mRNA targets of miR-371-3/302/367. The evidence of these miRNAs presence and overexpression in OGCT patients, decrease after surgery, and re-increase well-after resection is very well-established in the paper. However, miRNAs do not act alone to cause oncogenicity, and therefore, it would be further solidifying of the authors’ data to show inverse patterns of known targets of these miRNA such as LATS2, TGFBR2, CDKN1A, or PIK3R1 in each of the patients shown in Figures 4 and 5. Although the authors’ intention was not to provide evidence for a putative pathway, it would certainly add to the significance of their findings. To even further concretize the authors' findings, they could create a ceRNA network related to all miRNA in the paper to show possible avenues of action instead of simple correlations.

● Response: We appreciate the reviewer’s thoughtful input. Unfortunately, most of the tissue samples have been exhausted, preventing further analysis of downstream targets. As mentioned by the reviewer, the primary objective of this study is to confirm over-expression of these micro-RNA clusters also in ovarian germ cell tumors and that they can be detected in the plasma of patients with clinical evidence of non-teratoma germ cell tumors. These data could be therefore used to inform future studies that will confirm the diagnostic utility of circulating miRNAs in OGCT to help guiding clinical decision making in the postoperative setting and on surveillance.

2. Data-driven comparison of OGCT to TGCT.

a. The authors’ rely on the similarity of testicular germ cell tumors within the introduction, and discussion, sections but do not provide any comparative data. Considering that the TGCT are the more well-established tumor to miRNA dysregulation, a comparison to currently available data from these tumors would further solidify the connection between these primordial sex cells. Additionally, it could provide avenues for further investigation between the two to find similarities, differences, and correlations.

● Response: The current study has been inspired by the extensive data available in testicular germ cell tumors. Since there are many similarities between ovarian and testicular germ cell tumors in terms of treatments and outcomes, we wanted to explore if the same microRNAs were expressed also in OGCT, and if they were detectable in the peripheral blood. The corresponding author and many co-authors on this manuscript are involved in the clinical utility validation of miR371 in testicular germ cell tumors, leading large prospective studies (SWOG S1823). d. The largest and most clinically relevant studies in testicular germ cell tumors have been cited throughout the manuscript and discussed in the” discussion” paragraph. Our primary objective was not to compare OGCT with testicular GCT; it was to test whether or not these microRNAs were also expressed by OGCT and detectable in the peripheral blood. Any comparison between the OGCT and testicular GCT will be highly inaccurate and underpowered because of:

the heterogeneous patients’ populations. These miRNAs expression largely depend on clinical stage and tumor histology (not expressed in teratoma, lower expression in seminoma than in non seminoma). A comparison should be done by choosing matching testis cancer cases by stage, tumor subtypes and timing of blood collection (ex. prior to chemotherapy or surgery; post chemotherapy or surgery, etc..). This is unfortunately not feasible. Moreover, there will be no statistical power considering the very exiguous number of cases in each subgroup of the OGCT

3. Addressing tumor heterogeneity. a. Within the authors’ cohort, there is a wide-range of expression levels of mentioned miRNAs between patients (Figure 3). It would be helpful to delineate in a supplementary and separate figure which miRNAs are most overrepresented in specific tumor types as addressed in the discussion section. It’s one thing to recognize the difference in expression between tumor types, but it’s another, more useful, thing to present those differences quantitatively in graphical form. This could also add to the conclusion that germ cell tumors express specific miRNAs, regardless of sex type.

● Response: Thank you for your insightful comment regarding tumor heterogeneity and the variation in miRNA expression across patients. We acknowledge the wide range of expression levels observed in our cohort (Figure 3) and appreciate the suggestion to present these differences quantitatively for each tumor type. To address this, we have generated a supplementary figure (Fig S2) that provides a clearer quantitative comparison, of different miRNAs expression levels in different tumor types. Additionally, we have distinguished the tumor types using different colors in Figure 3 to enhance visualization.

Reviewer #3

1. LDH levels: Specify whether they are higher or lower than normal.

● Response: We clarified in the LDH level by changing “abnormal LDH” to “ above ULN” in Page 11 lines 225, 226

● 2. Line 212: Verify AFP data.

● Response: We verified the data and corrected the number in the text to 21, consistent with the table. Page 11 line 228

3. Use higher-resolution images for Figures 1 and 2.

● Response: Figures 1 and 2 have been replaced with higher-resolution versions to improve readability.

4. Clearly reference figure numbers and sub-figures in the Results section.

● Response: Figure and sub-figure numbers have been explicitly referenced throughout the Results section for clarity.

5. Conclusion should specifically point out the limitations of this study

● Response: We changed the last sentence of conclusion as follow, to point out the limitation. “Although these findings are based on a limited retrospective patient cohort, they suggest a potential role for miRNAs them as clinical biomarkers in OGCT”. Page 19 line 397-399.

---

## [Decision Letter · Decision Letter 1]

24 Mar 2025

Matching plasma and tissue miRNA expression analysis to detect viable ovarian germ cell tumors

PONE-D-24-42448R1

Dear Dr. Nappi,

We’re pleased to inform you that your manuscript has been judged scientifically suitable for publication and will be formally accepted for publication once it meets all outstanding technical requirements.

Kind regards,

Julie Decock, PhD

Academic Editor

PLOS ONE

Additional Editor Comments (optional):

Reviewers' comments:

Reviewer's Responses to Questions

**Comments to the Author**

1. If the authors have adequately addressed your comments raised in a previous round of review and you feel that this manuscript is now acceptable for publication, you may indicate that here to bypass the “Comments to the Author” section, enter your conflict of interest statement in the “Confidential to Editor” section, and submit your "Accept" recommendation.

Reviewer #2: All comments have been addressed

Reviewer #3: All comments have been addressed

2. Is the manuscript technically sound, and do the data support the conclusions?

Reviewer #2: Yes

Reviewer #3: Yes

3. Has the statistical analysis been performed appropriately and rigorously? 

Reviewer #2: Yes

Reviewer #3: Yes

4. Have the authors made all data underlying the findings in their manuscript fully available?

Reviewer #2: Yes

Reviewer #3: Yes

5. Is the manuscript presented in an intelligible fashion and written in standard English?

Reviewer #2: Yes

Reviewer #3: Yes

6. Review Comments to the Author

Reviewer #2: Thank you for your extensive answers. While I understand that you cannot test for mRNA targets due to tissue exhaustion, it is still vital to mention possible avenues of activity for the miRNAs themselves. Yes, there presence is correlated with these tumors but if you do not have a targeted pathway or gene, you have only found a correlation that may not be directly related to the causation of the tumor and could lead to false negatives/positives if used as a diagnostic. This is obviously more for the discussion section, however, future experimentation should require the testing of mRNA levels with miRNA in targeted cell types/in vivo.

Reviewer #3: uthors made significant efforts to address all the concern pointed by the reviewers.

I consider this manuscript is clearly written, well discussed with proper citation. So, I recommend the publication of this manuscript in the esteemed journal PLOS ONE.

7. PLOS authors have the option to publish the peer review history of their article (what does this mean? ). If published, this will include your full peer review and any attached files.

**Do you want your identity to be public for this peer review?** For information about this choice, including consent withdrawal, please see our Privacy Policy .

Reviewer #2: **Yes: ** Brian Glenn Jorgensen

Reviewer #3: No

---

## [Editor Report · Acceptance letter]

PONE-D-24-42448R1

PLOS ONE

Dear Dr. Nappi,

I'm pleased to inform you that your manuscript has been deemed suitable for publication in PLOS ONE. Congratulations! Your manuscript is now being handed over to our production team.

Kind regards,

on behalf of

Dr. Julie Decock

Academic Editor

PLOS ONE